

# Deletion of two-component system QseBC weakened virulence of *Glaesserella parasuis* in a murine acute infection model and adhesion to host cells

Xuefeng Yan[1,*], Ke Dai[2,*], Congwei Gu[1], Zehui Yu[1], Manli He[1], Wudian Xiao[1], Mingde Zhao[1] and Lvqin He[1]

[1] Southwest Medical University, Luzhou, China
[2] Sichuan Agricultural University, Chengdu, China
[*] These authors contributed equally to this work.

## ABSTRACT

The widespread two-component system (TCS), QseBC, involves vital virulence regulators in *Enterobacteriaceae* and *Pasteurellaceae*. Here we studied the function of QseBC in *Glaesserella parasuis*. A Δ*qseBC* mutant was constructed using a *Glaesserella parasuis* serovar 11 clinical strain SC1401 by natural transformation. Immunofluorescence was used to evaluate cellular adhesion, the levels of inflammation and apoptosis. The ability of Δ*qseBC* and Δ*qseC* mutant strains to adhere to PAM and MLE-12 cells was significantly reduced. Additionally, by focusing on the clinical signs, H&E, and IFA for inflammation and apoptosis, we found that the Δ*qseBC* mutant weakened virulence in the murine models. Together, these findings suggest that QseBC plays an important role in the virulence of *Glaesserella parasuis*.

## INTRODUCTION

*Glaesserella parasuis* is an opportunistic pathogen that causes Glässer's disease (GD); it is characterized by polyserositis, meningitis and arthritis, and has incurred significant economic losses in pig industries across many countries (*Oliveira & Pijoan, 2004*). To date, more than 15 serotypes have been found for this species (*Kielstein & Rappgabrielson, 1992*). It is difficult to prevent this disease by immunization of inactivated vaccines due to the lack of cross-protection among the different serotypes of *G. parasuis* (*Dai et al., 2019b*). Therefore, it is critical to elucidate the mechanism of *G. parasuis* survival and infection in the host for effective protective vaccine research.

*Glaesserella parasuis* encounters varying environmental stresses during its invasion process, which are vital for colonization and infection in the host (*Huang et al., 2016*; *Hughes et al., 2009*). Biofilms protect bacteria from elimination by the host immune system, which can enhance resistance of the bacteria to antibiotics and allow the exchange of genetic materials. Biofilms are considered to be the basis of many persistent chronic infections and the cause of repeated episodes of acute infection (*Kaplan & Mulks, 2005*; *Mah, 2012*).

Corresponding author
Lvqin He, 1176044269@qq.com

Biofilm positive strains of *G. parasuis* had positive correlation with resistance to β-lactams antibiotics (*Zhang et al., 2014*). Therefore, antibiotic resistance and biofilm-associated tolerance are important factors to consider in treating *G. parasuis* infections.

Quorum sensing (QS) is deemed as a special "language" for bacteria-bacteria communication in most bacteria such as *Vibrio fisheri*, *Pseudomonas aeruginosa*, *etc* (*Sperandio et al., 2003*). QseBC, a density-sensing regulator, plays an important role in the virulence of *Enterobacteriaceae* and *Pasteurellaceae* (*Weigel & Demuth, 2016*). Deletion of *qseC* in *Escherichia coli* and UroPathogenic *E. coli* leads to aberrant activation of QseB (*Kostakioti et al., 2009*). QseBC plays a vital role in the intracellular virulence of *Edwardsiella tarda* and *Actinobacillus pleuropneumoniae* (*Liu et al., 2015*; *Wang et al., 2011*). QseC is important for biofilm formation in *Aggregatibacter actinomycetemcomitans*, non-typeable *Haemophilus influenzae* and *Salmonella enterica* serovar Typhi (*Ji et al., 2017*; *Juarez-Rodriguez, Torres-Escobar & Demuth, 2013*; *Unal et al., 2012*). In mice infected with *Salmonella* serovar Typhimurium, deletion of *qseC* attenuated invasion of epithelial cells and systemic infections, and impaired survival within macrophages (*Bearson & Bearson, 2008*; *Moreira, Weinshenker & Sperandio, 2010*). However, the mechanism of QseB phosphorylation that regulates downstream genes requires further investigation.

In our previous studies, we found that CheY (QseB) plays vital roles in the growth and colonization of *G. parasuis*. QseC weakens the tolerance to stresses such as osmotic pressure, oxidative stress and heat shock. However, it was unclear whether QseBC was related to the bacterial virulence in vivo. In this study, we deleted the *qseB* and *qseC* genes in the wild-type strain of *G. parasuis* SC1401 and evaluated their abilities to adhere and invade cells. In addition, we also evaluated the reactive oxygen species inducing ability and inflammation/apoptosis levels in the murine models to analyze the virulence of the Δ*qseBC* mutant strain. This study aims to verify the role of QseBC and *G. parasuis* in the virulence of animals, which contributes to better understanding of the function of the two-component system and the mechanism that determines the virulence level of *G. parasuis*.

## MATERIALS AND METHODS

### Animals and ethics statement

Female BALB/c mice (6-week-old) were purchased from Chengdu Dossy Experimental Animal Co., Ltd (Sichuan, China). The animal experiments were conducted in strict accordance with the recommendations in the China Regulations for the Administration of Affairs Concerning Experimental Animals (1988) and had been approved by the Institutional Animal Care and Use Committee of Southwest Medical University (approval number 20211017-001), Luzhou, China. We did our best to provide maximum comfort and minimal stress for the mice.

### Bacteria strains, plasmids and culture conditions

Bacteria strains and plasmids employed in this study are listed in Table 1. *Escherichia coli* DH5α (Biomed, Beijing, China) was cultured in liquid Lysogeny Broth (LB, Difco, Thermo Fisher, USA) medium or on LB agar (Invitrogen, Shanghai, China) plates. *Glaesserella parasuis* strain SC1401 and its derivatives were grown in tryptic soy broth (TSB; Difco,
**Table 1  Bacteria strains and plasmids used in this study.**

| Strain or plasmid | Relevant characteristic(s) | Source |
|---|---|---|
| *G. parasuis* strains | | |
| SC1401 | Wild type, serovar 11 clinical isolate | Laboratory collection |
| △qseBC | SC1401 △qseBC::KanR | This study |
| C- △qseBC | SC1401 complemented △qseBC strain, Gm$^R$ Kan$^R$ | This study |
| *E. coli* strains | | |
| *E. coli* DH5α | Cloning host for maintaining the recombinant plasmids | Laboratory collection |
| *E. coli* BL21 | Expressing host for maintaining the recombinant plasmids | Laboratory collection |
| plasmids | | |
| pMD19-T | T-vector, Amp$^R$ | Takara |
| pET-28a | Expression vector, Kan$^R$ | Laboratory collection |
| pET-32a | Expression vector, Amp$^R$ | Laboratory collection |
| pk18mobsacB | Suicide and narrow-broad-host vector, Kan$^R$ | Laboratory collection |
| pLQ4 | A 2840-bp fragment containing Kan$^R$, the upstream sequences of *qseB* gene and downstream sequences of *qseC* gene in pK18mobsacB, Kan$^R$ | This study |
| PLQ5 | A 2056-bp fragment containing Gm$^R$ and the *qseBC* gene in PSF116 | This study |
| PSF116 | Gm resistance cassette-carrying vector, Gm$^R$ | *Zhou et al., 2016* |
| pKD4 | Amp$^R$, Kan$^R$, gene knock-out vector | Laboratory collection |

Notes.
 Kan, kanamycin; Gm, gentamicin; R, resistence.

NJ, USA) or on tryptic soy agar (TSA; Difco, Thermo Fisher, USA) supplemented with 0.001% nicotinamide adenine dinucleotide (NAD; Sigma-Aldrich, St Louis, MO) and 5% inactivated bovine serum (Solarbio, Beijing, China). When required, the medium was supplemented with kanamycin (kan; 100 μg/mL) or gentamicin (50 μg/mL).

## Construction and complementation of the *qseBC* mutant

The △*qseBC* strain and genetically complemented strains were constructed by natural transformation. Naturally competent bacteria are able to take up DNA fragments from the environment into their cytoplasm, and these foreign DNA fragments provide the bacteria with a source of nucleic acid (*Patrick & Melanie, 2013*). This transformation method is the third parallel gene transfer (HGT) mode, which is different from phage transduction and combined transfer, and is called natural transformation.

To construct an in-frame nonpolar *qseBC* mutant strain of *G. parasuis*, the target vector pLQ4 was built as follows: the upstream (958 bp) and downstream (947 bp) fragments of the in-situ *qseB* and *qseC* genes were amplified using primers P1+P2 (*qseBC*-Up-F/R, Table 2) and P3+P4 from the genomic DNA of SC1401. The *qseB* and *qseC* genes have a 14 bp overlap in *G. parasuis*. The kanamycin resistance cassette (935 bp) was amplified from pKD4 using primers P7+P8 (Kan-F/R). The three PCR fragments were integrated by overlap PCR with primers P1+P4 (*qseBC*-Up-F/*qseBC*-Down-R). The fusion segment was inserted into pk18mobsacB at the BamHI and HindIII sites to generate the recombinant plasmid pLQ4. Finally, the resulting plasmid pLQ4 was transformed into SC1401 using
**Table 2  Primers used in this study.**

| Primer name | Sequence (5′-3′) |
|---|---|
| For construction of mutant and complementary strain | |
| P1(*qseBC*-Up-F) | ctatgacatgattacgaattcACCGCTTGTAGCTACCCCTGAAGGCTTTAAAC |
| P2(*qseBC*-Up-R) | GCAGGGCTTCCCAACCTTACCTCAACCTCCCAATTTCAAAAAAG |
| P3(*qseBC*-Down-F) | GGGGTTCGAAATGACCGACCAGGATGGAGATATAAGGCACAAAC |
| P4(*qseBC*-Down-R) | caggtcgactctagaggatccACCGCTTGTCCGTCTTTAGTGATGGTTGGTG |
| P5(*qseBC*-Comp-F) | gaagtttctatgtaaggtaccAGATTACTGTTGATATTGATGACGAGG |
| P6(*qseBC*-Comp-R) | gcttatgtcaattcgggatcc TCAGGACGGAGTTTGACGGC |
| P7(*Kan*-F) | GTAAGGTTGGGAAGCCCTGCAAAGT |
| P8(*Kan*-R) | GGTCGGTCATTTCGAACCCCAGAGT |
| P9(*qseBC*-F) | ATGCGTATTTTATTAGTTGAAGACG |
| P10(*qseBC*-R) | TCAGGACGGAGTTTGACGGCAC |
| P11(HPS-F) | GTGATGAGGAAGGGTGGTGT |
| P12(HPS-R) | GGCTTCGTCACCCTCTGT |
| P13(*qseB*-F) | CGGGATCCATGCGTATTTTATTAATTGAAG |
| P14(*qseB*-R) | CCCAAGCTTTTAAGCAACTTCATCGTTTTTTC |
| P15(*qseC*-F) | CGGGATCCATGAAGTTGCTTAAAAATACC |
| P16(*qseC*-R) | CCCAAGCTTTCAGGACGGAGTTTGACGGC |
| P17(*qseBC* (200)-F) | AGATTACTGTTGATATTGATGACGAGG |
| P18(*qseBC* (200)-R) | CGTATTGCTCATTTTACCAAGTCAA |
| For RT-qPCR | |
| P19 A4U84_RS03675-F | GCGATTACTCCAGCAAGCCA |
| P20 A4U84_RS03675-R | ATCTCACCGCTTGCATCACG |
| P21 *groES*-F | GGCCATTGCCAACAGCAATC |
| P22 *groES*-R | TCGTCCGTTACACGACAAAGT |
| P23 16SrRNA-F | CCACCTGCCATAAGATGAGC |
| P24 16SrRNA-R | GGACCGTGTCTCAGTTCCAG |

a natural transformation technique. Transformant bacteria were incubated at 37 °C for 36 h. The kanamycin-resistant transformants were identified by PCR (using P7+P8 for the presence of Kan cassette, and P9+P10 for *qseBC* deletion). The target vector pLQ4 was built as follows: the fragment was ligated into a linearized vector PSF116, which was digested by Kpn I and BamHI using a BM seamless cloning kit (Biomed, Beijing, China). The pLQ5 was introduced into Δ*qseBC* by natural transformation. The complementary strain C-Δ*qseBC* was selected on TSA supplemented with 20 μg/mL gentamicin. The gentamicin-resistant transformants were identified by PCR. The flow chart for construction of the Δ*qseBC* mutant is shown in the attachment (Fig. S1).

To rule out the possible polar effects resulting from the deletion of *qseB* and *qseC* genes, relative quantification of 2 ($-\Delta\Delta C(T)$) method was adopted to analyze the transcriptional

levels of the flanking genes by quantitative real-time PCR (qRT-PCR) using primer sets P19/P20 for A4U84_RS03675 (hypothetical protein) and P21/22 for *groES* (co-chaperone), respectively. The stably transcriptional 16S RNA of *G. parasuis* was used as an internal reference gene. qRT-PCR was performed by using an iTaqTM universal SYBR Green Supermix (Bio-Rad, Hercules, CA) in a Lightcycler 96 (Roche, Switzerland) system.

## Preparation of 3D4/21 cell culture and adhesion and invasion assays

The study was performed as described in previous papers with some modifications (*Ji et al., 2017*; *Xu et al., 2015*; *Zhou et al., 2016*). The colony count method was used to study the adhesion and invasion ability of the Δ*qseBC* mutant strain to host cells. Porcine alveolar macrophages (PAM) 3D4/21 were purchased from Shanghai Zeye Biotechnology Co., Ltd. The 3D4/21 cells were cultured in Dulbecco's modified Eagle's medium (DMEM) supplemented with 10% (v/v) fetal bovine serum (FBS), 100 U/mL of penicillin G (Invitrogen, Waltham, MA), 100 mg/mL of streptomycin (Invitrogen, Waltham, MA) and MEM non-essential amino acids (Invitrogen, Waltham, MA). Cells were cultured at 37 °C in a humidified incubator with 5% $CO_2$ for 12 h before use. Then, cells at more than 80% confluence in 6-well plates were used for subsequent assays.

For the adhesion assay, the wild-type SC1401, derivatives of Δ*qseBC* and C-Δ*qseBC* were grown to logarithmic growth. These bacteria were washed in sterile PBS by centrifugation, resuspended in DMEM and adjusted by dilution to a multiplicity of infection (MOI) of 100:1 ($1 \times 10^7$ bacteria *vs.* $1 \times 10^5$ host cells). Plates were incubated at 37 ° C in 5% $CO_2$. The cells were incubated for 2 h for bacterial adhesion, and then vigorously washed three times with PBS to remove nonspecifically attached bacteria. Then the cells were incubated for 10 min at 37 °C with 100 μL 0.25% trypsin/0.02% EDTA. After incubation, 900 μL TSB was added, the cells were diluted with 10-fold serial for 4 times and 100 μL cultures were spread on TSA++ plates. The visible colonies were counted manually to calculate the colony forming units (CFU).

For the invasion assay, complete growth medium (including 150 μg/mL of gentamicin) was added to each well, and the plates were incubated for additional 2 h to kill extracellular *G. parasuis* and washed three times with PBS. The above cells were lysed with 1% Triton X-100 to determine the intracellular bacteria by serial dilutions. The visible colonies were counted manually to calculate the CFU. The assay was performed in triplicates.

## Indirect immunofluorescent assay of adherent bacteria on MLE-12 cells

Mouse lung epithelial cells MLE-12 were purchased from the Beijing Institute of BeNa Biotechnology (ATCC-sourced strain, the 4th passage). In the indirect immunofluorescent (IIF) assay, MLE-12 cells were grown on coverslips in a six-well plate to reach a cell concentration of 90% confluence. The bacteria and cells were co-incubated following the adhesion assay. In this study, the *G. parasuis* HpGbpA polyclonal antibody (1:500 dil.) prepared and stored in our laboratory was used as the primary antibody, and the commercial CY3-labeled goat anti-mouse IgG (Servicebio, Gent, Belgium) was used as the secondary antibody to detect the distribution of bacteria. Cy3-labeled fluorescent secondary antibodies

(EX WL: 510–560 nm, EM WL: 590 nm) were detected by a fluorescence microscope Nikon Eclipse ci&NIKON DS-U3, and the fluorescence intensity was analyzed by CaseViewer software. Fluorescence signal intensity was analyzed using ImageProPlus software and expressed as integrated optical density (IOD). The relative IOD value ($IOD_R$) is the ratio of the fluorescence intensity of CY3 ($IOD_C$) to the fluorescence intensity of DAPI ($IOD_D$).

## Animal infection studies

Animal infection studies were performed as described previously (*Dai et al., 2019a*; *Dai et al., 2021*; *Ding et al., 2016*; *Zhang et al., 2016*). All efforts were made to provide for maximum comfort and minimal suffering of the animals. The study was conducted in accordance with ethic guidelines of the recommendations in the China Regulations for the Administration of Affairs Concerning Experimental Animals (1988) and the study was approved by the Institutional Animal Care and Use Committee of Southwest Medical University (20211017-001), Sichuan, China.

Thirty-two mice were allocated randomly to four groups of eight. Three groups were intraperitoneally injected with the *G. parasuis* SC1401, △*qseBC* or C-△*qseBC* strain at a dose of 1. 3 × $10^8$ CFU per mouse. The remaining group was allocated to the PBS control group and injected intraperitoneally with 0.5 mL PBS. Dying animals that exhibited lethargy, hunched backs, rough coats, swollen abdomens, or inability to eat or drink were euthanized and considered dead.

Random block grouping method: mice were allocated randomly to SC1401, △*qseBC*, C-△*qseBC* and mock groups, with each group containing eight randomized mice. The random number table was then used to randomly assign the animals to the treatment group and the control group for experimental observation. In order to reduce the differences in experiments for non-research purposes, when the animals were injected intraperitoneally, groups were randomly selected, and a new set of syringes and needles was used for each individual animal. The animal cages were placed on the same level in the cage rack. In this blinded experiment, animal management personnel were not informed about the animal groupings, and the observations were done without information on the grouping of animals, so as to reduce the artificial selection bias in the observation of the results.

All mice were intraperitoneally injected with bacteria at a dose of 1.3 × $10^8$ CFU/animal (0.5 mL) or with 0.5 mL PBS. Clinical symptoms were carefully monitored for 7 days. Euthanasia was carried out on the animals with sleepiness, hunchback posture, rough hair, abdominal distension, or inability to eat or drink. The method of euthanasia was intraperitoneal injection of sodium pentobarbital (150 mg/kg). After 72 h of challenge, three animals in each group were euthanized humanely. The spleens and lungs were autopsied and treated with formalin/paraformaldehyde-fixed, paraffin-embedded (FFPE), and used for hematoxylin eosin (H&E) analysis to determine the pathological damage before and after immunization. The surviving mice were humanely euthanized (7 dpi).

## Histological examination (H&E) and immunofluorescence assay

Mice were euthanized using sodium pentobarbitone (150 mg/kg). Lung and spleen tissues were harvested from mice in different groups (SC1401, △*qseBC*, C-△*qseBC* and PBS) after

4 dpi and fixed with 4% paraformaldehyde for histopathological and immunofluorescence (IF) assay. The paraffin sections were deparaffinized to water, and the sections were stained with hematoxylin and eosin (H&E) in turn, dehydrated, sealed with neutral gum and examined under a microscope (*Dai et al., 2021*).

Myeloperoxidase (MPO) levels were measured through the IF assay to quantitatively evaluate the levels of inflammation in the lungs of mice infected with SC1401, △qseBC and △qseC. The experimental procedure is described in *Dai et al. (2019a)*. Mouse tissue and partial preparations were similar to the preparation of H&E materials. The fluorescent secondary antibody labeled with CY3 (EX WL: 510–560 nm, EM WL: 590 nm) was detected by a fluorescent microscope Nikon Eclipse ci&NIKON DS-U3, and the fluorescence intensity was analyzed by CaseViewer software. Fluorescence signal intensity was analyzed using ImageProPlus software and expressed as integrated optical density (IOD). The relative IOD value ($IOD_R$) is the ratio of the fluorescence intensity of CY3 ($IOD_C$) to the fluorescence intensity of DAPI ($IOD_D$).

### Determination of the level of apoptosis in lung tissue (TUNEL staining)

In order to study cell apoptosis *in situ* in mice, the lungs of mice in the SC1401, △qseBC and △qseC challenge groups were collected, and then the TUNEL (TdT-mediated dUTP nick end labeling) kit was used to detect the level of apoptosis (*Dai et al., 2019a*). Fluorescence microscope Nikon Eclipse ci&NIKON DS-U3 was used to detect FITC-labeled fluorescent secondary antibody (EX WL: 465–495 nm, EM WL: 515–555 nm), and the fluorescence intensity was analyzed by CaseViewer software. Fluorescence signal intensity was analyzed using ImageProPlus software and expressed as integrated optical density (IOD). The relative IOD value ($IOD_R$) is the ratio of the fluorescence intensity of FITC ($IOD_F$) to the fluorescence intensity of DAPI ($IOD_D$).

### Statistical analyses

Statistical analyses were performed using Graph-Pad Prism 5.0 software (GraphPad Software, San Diego, CA, USA). A one-way or two-way ANOVA analysis was used to compare the differences among more than two groups. *P* values < 0.05 were considered statistically significant.

## RESULTS

### The *qseB* and *qseC* genes are conserved among 15 standard strains of *G. parasuis*

The standard strains of *G. parasuis* 1–15 were used as the templates to amplify the *qseB* and *qseC* genes by PCR, and the target bands were obtained (*qseB* is 672 bp; *qseC* is 1,398 bp). It was shown that the 15 standard strains of *G. parasuis* serotype all have *qseB* and *qseC* genes (Figs. 1A/1B).

### Validation of the *qseB/C* genes deletion and complementation

The recombinant plasmid pLQ4 was identified by restriction enzyme digestion (Fig. 2A). The recombinant plasmid pLQ4 was digested into a vector fragment (5,600 bp) and an
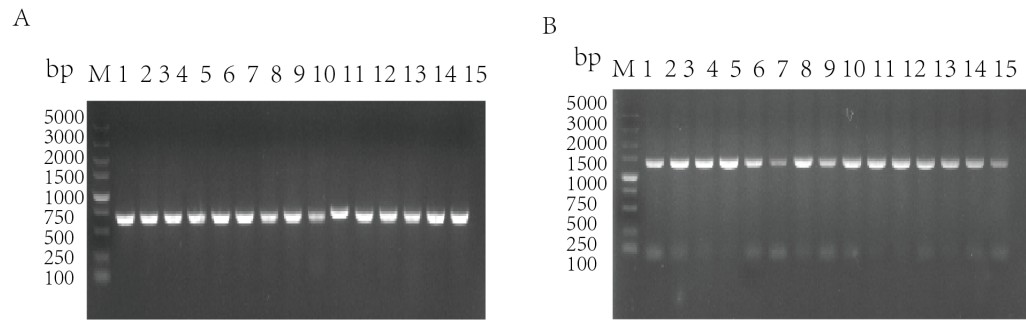

**Figure 1 Amplification of *qseB* gene (A) and *qseC* gene (B) from serotypes 1–15 of *G. parasuis* standard strains.** M: DL5000 DNA marker; 1–15: Serotypes 1–15 of *G. parasuis* standard strains.

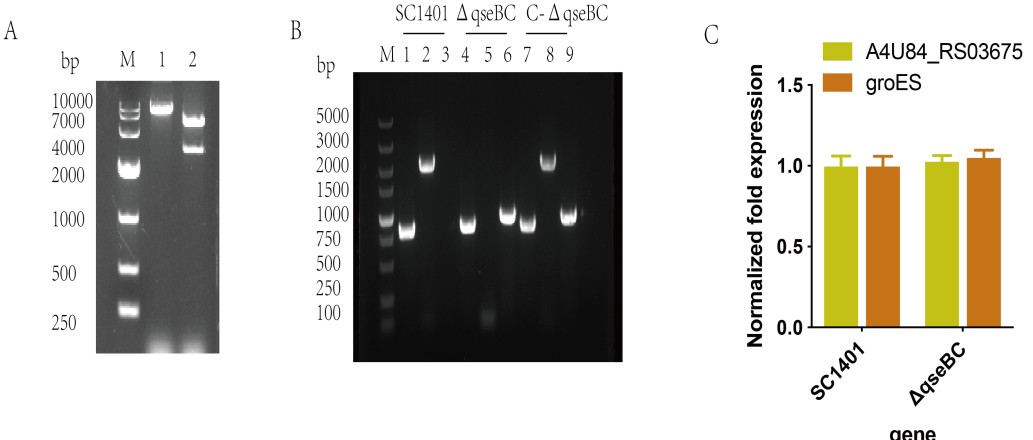

**Figure 2 Construction and verification of the *qseBC* mutant and the complemented strain (*qseB* and *qseC* have a 14-bp overlap).** (A) Identification of the recombinant plasmid pLQ4 (lane 1: BamHI; lane 2: EcoR I and BamH I) by restriction enzyme digestion. (B) PCR identification of strains SC1401, Δ*qseBC* and C-Δ*qseBC*. Lane 1, 4, 7: primers HPS-F/HPS-R amplify HPS 16S rRNA gene; lane 2, 5, 8: primers *qseBC*-F/*qseBC*–R amplify *qseB* and *qseC* genes; lane 3, 6, 9: primers Kan-F/Kan-R amplify the kan gene. (C) RT-qPCR verification of polarity effect. Transcription levels of the neighbouring genes in the Δ*qseBC* mutant were determined by RT-qPCR using 16S rRNA as an internal control.

insert fragment (2,840 bp). The in-frame nonpolar Δ*qseBC* mutant was constructed by natural transformation and confirmed by PCR (Fig. 2B). The identities of SC1401, Δ*qseBC* and C-Δ*qseBC* were confirmed in all strains by amplifying the 822-bp fragment encoding 16S rRNA using primers P11/P12. A 2061-bp fragment of the *qseBC* gene using primers P9/P10 was amplified from the wild type and complement, but not the mutant. The kan fragment (935 bp) was amplified using primers P7/P8 from the mutant and complement, but not from the wild-type strain SC1401, indicating that the gene mutant and its complemented strain were successfully constructed.

The transcription of the *qseB* and *qseC* flanking genes were further examined by RT-qPCR to assess the possible polar effects caused by deletion of the *qseB* and *qseC* genes. As

shown in SC1401 and $\Delta qseBC$, both the upstream gene A4U84_RS03675 and downstream gene *groES* were not significantly affected (Fig. 2C), indicating no polar effect.

### The $\Delta$*qseBC* mutant decreased adherence and invasion abilities

In order to further study the effect of *G. parasuis* two-component system QseBC on the interaction between bacteria and host cells, the PAM cells were co-incubated with SC1401, $\Delta qseBC$ and $\Delta qseC$ to compare their adhesion and invasion abilities. As shown in Figs. 3A/3B, using bacteria in the same growth state (taking $OD_{600}$nm as a reference) and the same batch of plated cells, when MOI =100, the average number of cells adhered to PAM cells by SC1401, $\Delta qseBC$ and $\Delta qseC$ were 311.67 $\times 10^5$ CFU/well, 178.67 $\times 10^5$ CFU/well and 223.33 $\times 10^5$ CFU/well, respectively. As compared to the parent strain, both mutant strains showed a significant decrease. In addition to the decreased number of adhesions, deletion of the *qseB/C* genes also significant impaired the ability of the bacteria to invade the PAM cells ($p < 0.05$). The adhesion and invasion levels were recovered in the complemented strain C-$\Delta qseBC$.

In addition, the decreased adhesion of $\Delta qseBC$ and $\Delta qseC$ to mouse-derived MLE-12 cells was observed using the IIF assay. The HpGbpA polyclonal antibody (1:500 dil.) prepared and stored in our laboratory was used as the primary antibody, and the commercial Cy3-labeled goat anti-mouse IgG (Servicebio, Gent, Belgium) was used as the secondary antibody to climb the slides of MLE-12 cells. Indirect immunofluorescence assay was performed. The DAPI (blue) column represents the nuclear staining results; the CY3 column (red) represents the bacterial staining results; the third column is the combined graph of the two (observing the results of bacterial adherent cells). Fluorescence signal intensities are displayed as integrated optical density (IOD) values and analyzed by ImageProPlus software. The relative value of $IOD_R$ was calculated by the ratio of Cy3-fluorescence intensity ($IOD_C$) and IOD of DAPI ($IOD_D$). As shown in Figs. 3C/3D, the relative fluorescence intensity (RFI) of the deletion strain was significantly lower than that of the parent strain, and the red fluorescence was localized on the cell surface, which was consistent with the location of bacteria adhering to the cell surface. The results of this part further confirmed that the QseBC two-component system was involved in the regulation of bacterial adhesion to cells.

In summary, our results indicated that $\Delta qseBC$ resulted in a significant decrease in the adhesion and invasion abilities of the bacteria to the PAM and MLE-12 cells used in the experiment, suggesting that QseBC affects the ability of the bacteria to interact with PAM and MLE-12 cells.

### Decreased virulence of the $\Delta$*qseBC* mutant in mice

The SC1401 and the $\Delta qseC$ or $\Delta qseBC$ strain were compared in a murine model of infection. After intraperitoneal injection with SC1401, the survival rate was 12.5% (1 mouse), whereas the survival rate was 50% (4 mice) in both the $\Delta qseBC$ and $\Delta qseC$ groups. No anomalies were observed in the PBS control group throughout the entire virulence assays. In short, 87.5% of mice inoculated with SC1401 died as compared to only 50% of mice inoculated with $\Delta qseBC$ and $\Delta qseC$, respectively (Fig. 4A), demonstrating that QseBC contributes to the virulence of *G. parasuis*.

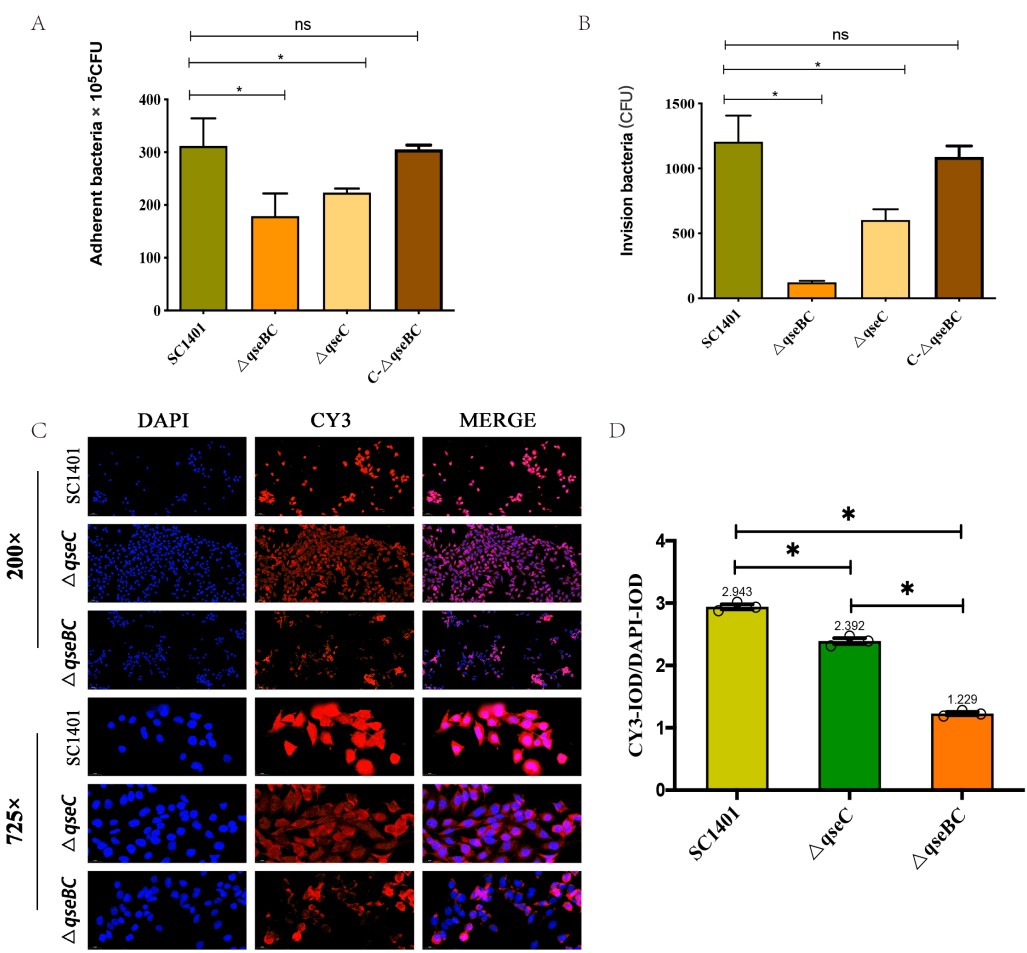

**Figure 3** **Adhesion and invasion abilities of *G. parasuis* to PAM and MLE-12 cells.** (A) Adhesion to PAM cells. (B) Invasion to PAM cells. (C) Indirect immunofluorescent analysis (IIF) of adherent Δ*qseBC* and Δ*qseC* mutants to MLE-12 cells. IIF assays were conducted to detect the distribution of bacteria by mouse-derived HpGbpA polyclonal antibody (1:500 dil.) and Cy3-conjugated goat-anti-mouse IgG (Servicebio). (D) Statistical analysis of IOD of MLE-12 cells. Relative values of $IOD_R$ were calculated by the ratio of the IOD of CY3-fluorescence intensity ($IOD_C$) and DAPI ($IOD_D$). The experiments were performed three times independently in triplicates. Error bars represent the standard errors from three independent experiments.

Lung and spleen tissues of hosts were harvested for histological pathological observation using H&E staining. As shown in Figs. 4B/4C, pathological lesions were present in all treated groups. The mice in wild-type and complemented groups showed severe congestion of alveolar wall capillaries. However, mice in Δ*qseC* and Δ*qseBC* groups demonstrated less severe pathological changes.

## MPO immunofluorescence analysis

In this experiment, through the IF analysis of the lung pathological tissue of the infected mice, it was found that 4 days after infection (4 dpi), the IOD of the lung tissue of the mice in each treatment group was significantly higher than that of the blank group ($p < 0.05$),

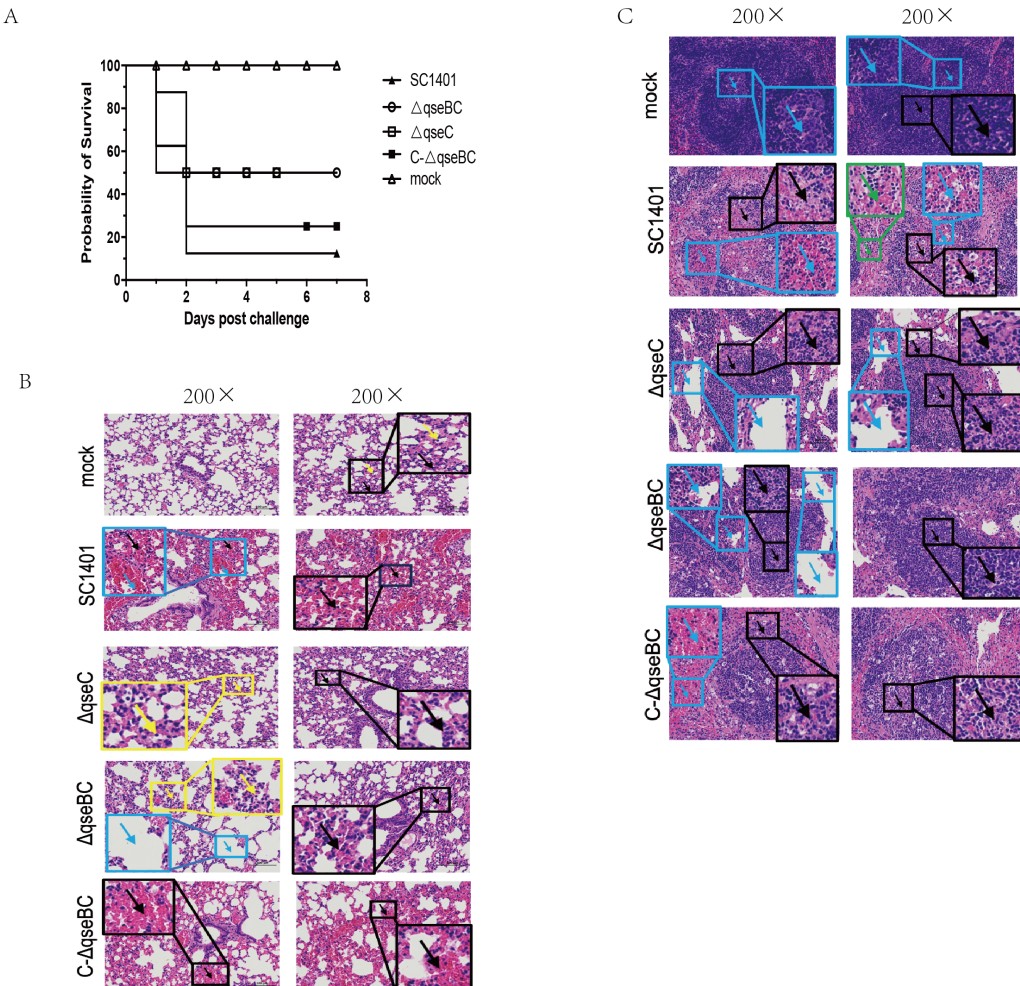

**Figure 4** **Survival curves of mice challenged with *G. parasuis* and histopathologic analysis of lungs and spleens (200×).** (A) Survival curves of mice infected intraperitoneally with SC1401, △*qseBC*, △*qseC* or C-△*qseBC* strains. The survival percentage of mice infected with △*qseBC* or △*qseC* was significantly higher than that of the wild-type strain using the log-rank test ($P < 0.05$). (B) Lung tissues of mice challenged with *G. parasuis*. **Mock group** (Mock): There was no obvious abnormality in the morphology and structure of the airway; a small amount of alveolar wall thickening (black arrow) with scattered inflammatory cell infiltration (yellow arrow) can be seen. **Wild-type group** (SC1401): There was no obvious abnormality in the morphology and structure of the airway; extensive alveolar wall capillary hyperemia (black arrow) and narrow alveolar cavity can be seen; vascular congestion was more common (blue arrow); no obvious inflammatory cell infiltration seen. ***qseC* mutant group** (△*qseC*): There was no obvious abnormality in the morphology and structure of the airway; a large area of the alveolar wall was slightly thickened (black arrow), accompanied by scattered inflammatory cell infiltration (yellow arrow). ***qseBC* mutant group** (△*qseBC*): There was no obvious abnormality in the morphology and structure of the tissue airway; more alveolar walls were slightly thickened (black arrows), and some alveolar cavities were narrowed with scattered inflammatory cell infiltration (yellow arrows); a small number of alveolar sacs were slightly expanded (Blue arrow). ***qseBC* complemented group** (C-△*qseBC*): There was no obvious abnormality in the morphology and structure of the airway; more severe congestion of the alveolar wall capillaries (black arrow) was seen; no obvious inflammatory cell infiltration seen. (continued on next page...)

**Figure 4 (…continued)**
(C) Spleen tissues of mice challenged with *G. parasuis*. **Mock group** (Mock): Abundant white pulp was seen in the tissues, some white pulps were enlarged in volume and some had irregular shapes (black arrows); a few germinal centers (blue arrows) were seen in the white pulp; sinus dilation observed. **Wild-type group** (SC1401): The white pulp of the tissue was severely damaged and the number of lymphocytes was greatly reduced, with only a few remaining present; severe lymphoid necrosis (black arrow), nucle i fragmentation, pyknosis, disappearance, and bleeding of the white pulp (blue arrow) were observed; red; pulp congestion and bleeding seen (green arrow). ***qseC* mutant group** ($\Delta qseC$): The tissue white pulp and red pulp cells were plasmacytoid lymphocytes; the white pulp was moderately damaged and the number of lymphocytes was reduced; moderate lymphoid necrosis (black arrow) and nuclei fragmentation can be seen; a large number of red pulp sinus dilatation (blue Arrow) in a honeycomb shape was seen. ***qseBC* mutant group** ($\Delta qseBC$): The tissues of the white pulp and red pulp cells were plasmacytoid lymphocytes; the white pulp was slightly injured and the number of lymphocytes was reduced; some lymphoid necrosis (black arrow) and nuclei fragmentation was seen; a large amount of red pulp sinus expansion (blue arrow), in a honeycomb shape was seen. ***qseBC* complemented group** (C-$\Delta qseBC$): The white pulp of the tissue was severely damaged and the lymphocytes were significantly reduced, with only a few lymphocytes remaining; severe lymphoid necrosis (black arrow), nuclei fragmentation, pyknosis, and disappearance was seen; red pulp congestion and hemorrhage (blue arrow).

suggesting that the activity of MPO was increased. The $IOD_R$ of SC1401 (average $IOD_R$: 0.625) and complemented groups (average $IOD_R$: 0.582) were significantly higher than that of the $\Delta qseBC$ mutant group (average $IOD_R$: 0.190) and $\Delta qseC$ mutant group (average $IOD_R$: 0.371), indicating that $\Delta qseBC$ and $\Delta qseC$ mutants caused significantly lower secondary inflammatory active substances in the lungs of mice than the parent strain, and the inflammatory damage to the target tissues may be more minor (Fig. 5). The evaluation results of the above pathological sections were further confirmed.

## Analysis of tissue apoptosis level in challenged mice

In order to directly observe the pathological damage at the level of cell death, the level of bacterial-induced apoptosis in mouse lungs was quantitatively analyzed by TUNEL staining. As shown in Fig. 6, the IOD of the FITC-labeled fluorescent secondary antibody was significantly enhanced in all treatment groups. Taking the $IOD_D$ of DAPI in each group as a reference, the average $IOD_R$ values of mice in the $\Delta qseBC$ and $\Delta qseC$ challenge groups were only 18.88% and 16.78% of the parental strain challenge groups, respectively. The mean $IOD_R$ of $\Delta qseBC$ and $\Delta qseC$ were 0.027 and 0.024, respectively, while the $IOD_R$ of the wild-type SC1401 group was 0.143. Replenishing strain group can compensate for a certain degree of cell damage ability (average $IOD_R = 0.098$). However, the connection between QseBC and some apoptosis-mediated pathways is uncertain and needs to be further studied in the future.

## DISCUSSION

*Glaesserella parasuis* (*G. parasuis*), a common pathogenic bacterium in the upper respiratory tract of pigs (*Zhao et al., 2018*), is currently one of the main bacterial pathogens harming most of the pig industry in China and even across the world. The currently reported *G. parasuis* virulence factors mainly include: bacterial adhesion factors (outer membrane proteins OmpA/OmpP5 and OmpP2, fimbriae system PilABCD, *etc.*) (*Varela et al., 2014*), self-regulation and survival-related factors (binary transduction system ArcAB, QseBC,

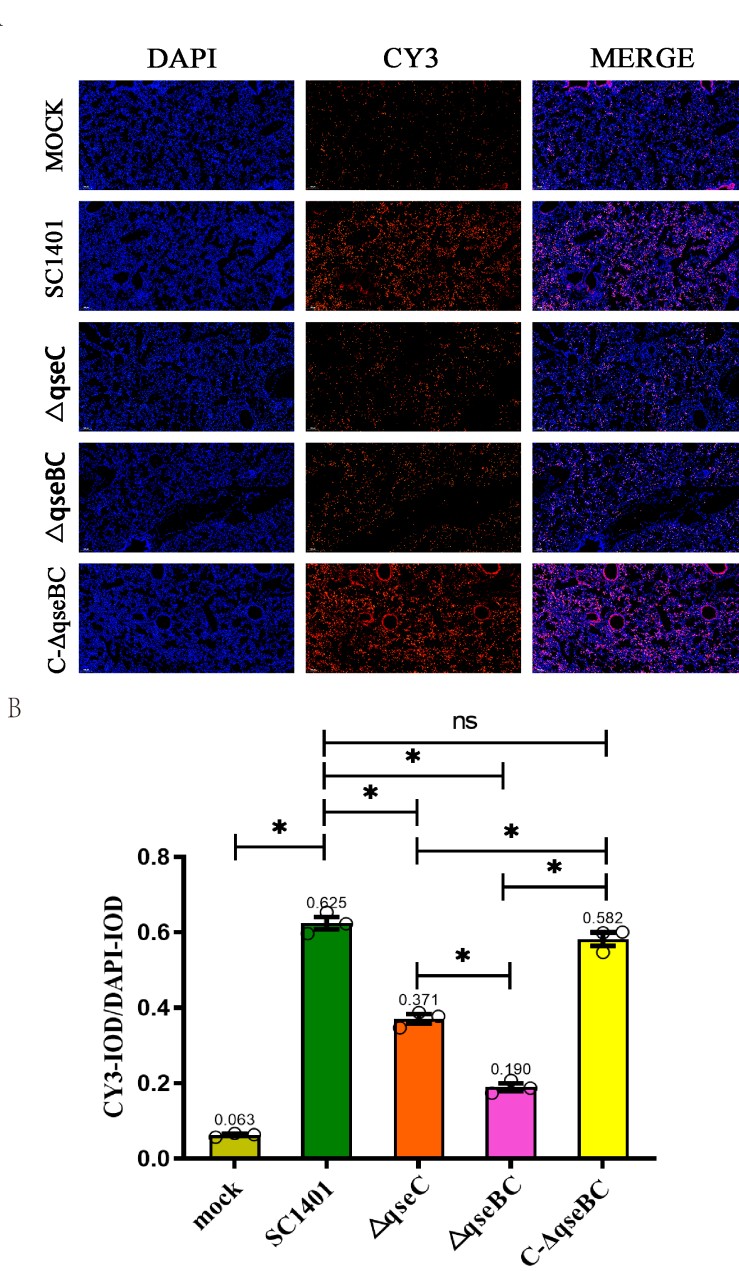

**Figure 5** **Immunofluorescent (IF) assay for the level of myeloperoxidase (MPO) in l ung tissues.** Lung tissues were harvested from mice in different groups after 4 dpi and used for FFPE treatment, pathological slice preparation, and IF assay for MPO. (A) IF analysis of lung tissue slides (100×). (B) Statistical analysis of IOD of myeloid cells (Biomarker: MPO) in mice lungs. Relative values of $IOD_R$ were calculated by the ratio of the IOD of CY3-fluorescence intensity ($IOD_C$) and DAPI ($IOD_D$). T he experiments were performed three times independently in triplicates. Error bars represent the standard errors from three independent experiments.

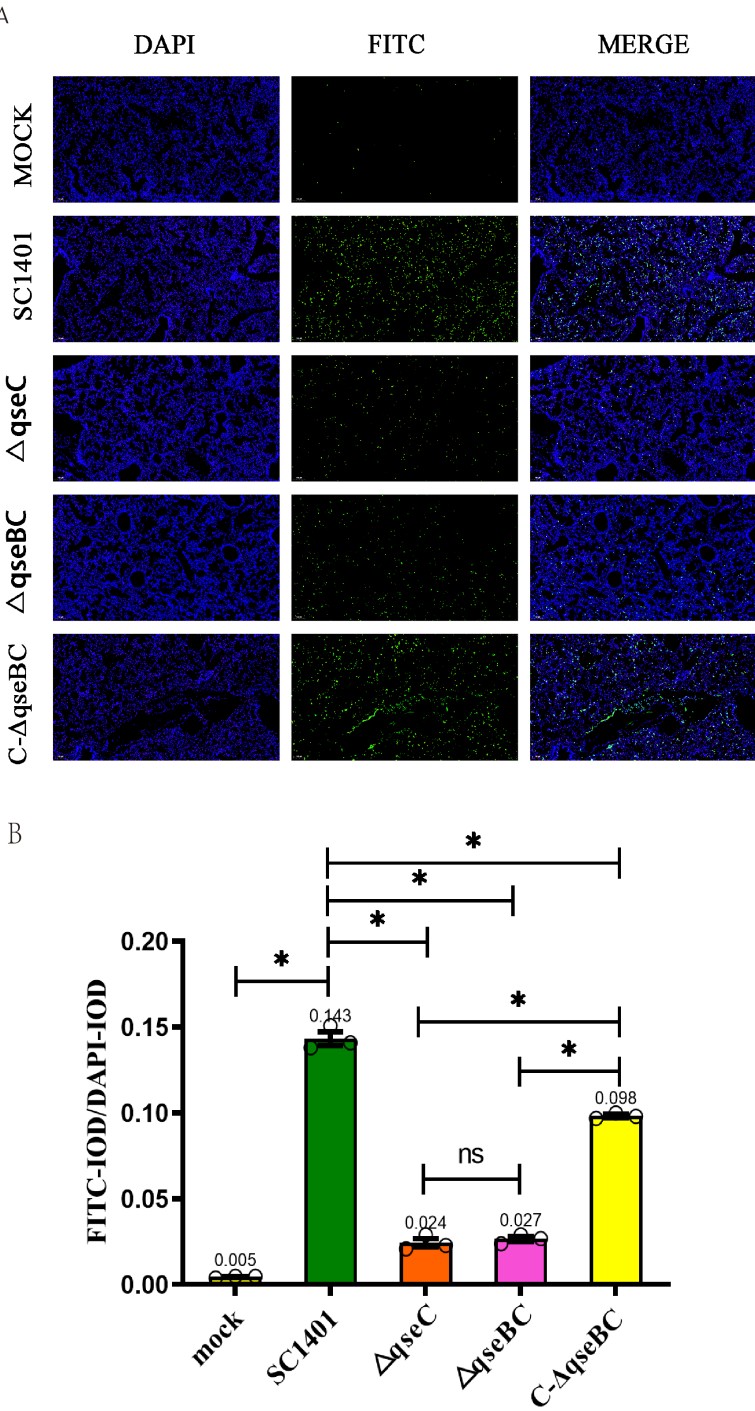

**Figure 6    Apoptosis assay for the level of pathological lesion in lung tissues (TUNEL staining).** Lung tissues were harvested from mice in different groups after 4 dpi and used for FFPE treatment, pathological slice preparation, and IF assay for apoptosis. (A) IF analysis (100×). (B) Statistical analysis of IOD of myeloid cells (Biomarker: mpo) in mice lungs. Relative values of $IOD_R$ were calculated by the ratio of the IOD of FITC-fluorescence intensity ($IOD_F$) and DAPI ($IOD_D$).

*etc.*) (*Ding et al., 2016*; *He et al., 2018*; *He et al., 2016*), bacterial invasion related factors (collagenase PrtC, brain tissue invasion related SphB), bacterial immune evasion factors (GalU and GalE) (*Zou et al., 2013*), bacterial iron synthesis and uptake proteins (transferrin receptor Tbp) (*Gutiérrez-Martín, 2016*), and bacterial toxins (*Li et al., 2017*). Among them, bacterial toxins, mainly including lipopolysaccharide/endotoxin (LPS) (*Perry et al., 2013*) and cell lethal expansion toxin (CDT family) (*Zhang et al., 2012b*), are the direct virulence factors of *G. parasuis*, as well as the most important pathogenic factors of *G. parasuis* (*Li et al., 2017*; *Perry et al., 2013*). The two-component system QseBC is a ubiquitous signal transmission and gene expression regulation system in bacteria. It participates in the regulation of bacterial growth and reproduction, metabolism, motility, drug resistance and virulence factor expression (*Appleby, Parkinson & Bourret, 1996*). Inoculation of commercial inactivated vaccines is the main measure to prevent and treat *G. parasuis* infection. However, there are many serotypes of *G. parasuis*, and inactivated vaccines will only produce specific antibodies against lipooligosaccharides of the same serotype. Insufficient cross-reactivity makes inactivated vaccines of a single serotype an ineffective means of protection (*Dai et al., 2021*; *Mccaig et al., 2016*). Therefore, understanding the molecular mechanism of pathogenicity is essential for the development and targeted therapy of new *G. parasuis* vaccines.

The partners of different two-component systems showed significant cross specificity in *Escherichia coli*. Absence of the quorum-sensing *E. coli* (Qse)BC sensor QseC results in a strong cross interaction between its homologous partner QseB and the polymyxin resistance two-component system PmrAB (*Guckesa et al., 2013*). In the absence of homologous sensors, this cross interaction is harmful and seriously weakens the virulence of pathogens. In previous studies, we have constructed the Δ*qseB* (Δ*cheY*) and Δ*qseC* mutant strains, and investigated the functions of QseB and QseC in *G. parasuis*. We found that the Δ*qseB* mutant strain demonstrated slower growth rate, less biofilms formation and slower agglutination compared with the wild-type strain. However, there were no observed apparent differences of serum-resistance between the wild-type and gene-mutant strain. Moreover, the Δ*qseC* exhibited a decreased resistance to osmotic pressure, oxidative stress and heat shock. We also found that the Δ*qseC* had weakened the ability to take up iron and showed defective biofilm formation, and that QseC participated in sensing the epinephrine in environment to regulate the density of *G. parasuis*. However, the entire two-component system QseBC knockout in *G. parasuis* had not been reported. These results encouraged us to study the basic molecular functions of this important signal transduction system in *G. parasuis*. It is still unclear whether the decreased virulence of Δ*qseC* mutant strain is related to the activation of QseB by PmrB, thus requiring further study. It is also not clear whether there is strong crosstalk from other two-component systems in Δ*qseB* and Δ*qseC* mutant strains. Therefore, we knocked out *qseB* and *qseC* double genes to elucidate the role of QseBC in the pathogenicity of *G. parasuis* and provide the basis for further study on the pathogenic mechanism of *G. parasuis*.

*Glaesserella parasuis* can invade porcine brain microvascular epithelial cells (PBMEC), which are the main cells in the blood–brain barrier (BBB), indicating that *G. parasuis* can cross the BBB to reach the central nervous system. Previous studies have found that *G.*
*parasuis* has the ability to adhere and invade a variety of cells, such as porcine umbilical artery endothelial cells (PUVEC), porcine kidney epithelial cells (PK-15), alveolar macrophages (PAM) cells, *etc.* (*Bouchet et al., 2008*; *Zhang et al., 2012a*; *Zhou et al., 2012*). However, more in-depth research is needed to elucidate the molecular mechanism by which *G. parasuis* breaks through the mucosal barrier. In this study, we found that deletion of the *qseBC* gene resulted in a significant decrease in the ability of *G. parasuis* to adhere to host cells. It was confirmed that similar to *Actinobacillus pleuropneumoniae*, the two-component system QseBC in *G. parasuis* may regulate bacterial adhesion by regulating the expression of downstream virulence genes (*Liu et al., 2015*). Based on the above results, we hypothesize that the loss of *qseBC* gene expression leads to a decrease in the ability of the pathogenic bacteria to adhere and invade host cells, which may be part of the reason for the weakened virulence of *G. parasuis.*

Animal infection studies in this study further assess the importance of the QseBC two-component system. We combined our findings in this study with our previous study that showed that the QseBC two-component system functions as a global virulence regulator (*Weigel & Demuth, 2016*). We verified that deletion of *qseBC* genes in *G. parasuis* can lead to attenuated virulence in the BALB/c mouse model. Mice infected with the Δ*qseBC* mutant had a 37.5% reduction in mortality as compared to mice infected with SC1401 (Fig. 4). This is similar to the results of *qseB* or *qseC* gene deletion in *Haemophilus influenzae* (*Wang et al., 2011*). In order to understand how the lack of QseBC affects the pathogenicity of *G. parasuis*, we collected mouse lungs and spleens for pathological sectioning, observed and compared the pathological changes, and found that the Δ*qseBC* mutant caused low fatal infection rate and relatively mild pathological damage. The alveolar wall was slightly thickened with scattered inflammatory cell infiltration in the Δ*qseC* group. In the Δ*qseBC* group, more alveolar walls were found to be slightly thickened, and some alveolar cavities were narrowed with scattered inflammatory cell infiltration, with a small number of alveolar sacs observed to be slightly expanded (Fig. 4B). In the spleen tissues of mice in wild-type groups, the white pulp was severely damaged and displayed dilatation, and the number of lymphocytes was greatly reduced, with only a few lymphocytes remaining present. Moreover, severe lymphoid necrosis, nuclei fragmentation, pyknosis, disappearance, and bleeding of the white pulp in the tissues were observed. However, in the Δ*qseC* group, the tissues of the white pulp and red pulp cells were plasma cell-like lymphocytes; white pulp was moderately injured, with reduced number of lymphocytes, and more lymphoid necrosis and nuclear fragmentation were seen; a large number of red pulp sinuses were expanded in a honeycomb pattern. In the Δ*qseBC* group, the tissues of the white pulp and red pulp cells were plasmacytoid lymphocytes. Moreover, we observed reduced number of lymphocytes, and slight lymphoid necrosis and nuclei fragmentation in the tissues (Fig. 4C). No significant bleeding and dilation of the spinal sinuses were observed in the mock group. This data showed that while the Δ*qseC* and Δ*qseBC* mutants could still cause certain degree of pathological lesion, the symptoms and pathological changes were less pronounced in mutant groups than wild-type and complemented groups, indicating that the deletion of *qseB* and *qseC* genes led to a significant attenuation of the degree of pathological damage caused by *G. parasuis* in the mouse model.

Indirect immunofluorescence assays were performed on mouse lungs to compare myeloperoxidase (MPO) levels to assess the development of inflammation. IF analysis showed that the Δ*qseBC* mutant produced lower levels of oxidative damage and less severe levels of inflammation and apoptosis. In order to directly observe the pathological damage at the level of cell death, the level of bacterial-induced apoptosis in mouse lungs was quantitatively analyzed by TUNEL staining. We confirmed that the rate of apoptosis induced by the Δ*qseBC* and Δ*qseC* mutant strains was significantly reduced as compared to the wild strain. In conclusion, our results show that Δ*qseBC* has weak pathogenicity in mouse models, and its abilities to induce inflammation and apoptosis are significantly lower than that of parental and apoplectic strains for mice treated with the same dose of bacteria. After undergoing the challenge, mice in the mutant strain group had weaker pathological manifestations and lower mortality. This indicates that the QseBC two-component system can significantly regulate the virulence of HPS to the host. Subsequent research can use transcriptomes, proteomes, combined with gene regulation/interaction network, and other means to comprehensively study the regulatory mechanism behind it.

## CONCLUSIONS

In conclusion, our results indicate that QseBC is essential for the bacterial virulence of *G. parasuis* in the mouse model. A key uncertainty factor is that the mouse model cannot reflect the exact infection process of *G. parasuis* in its natural host. The pig should be used as a model to further evaluate the virulence of these strains. Our research can lay the foundation for the follow-up study of the virulence of QseBC in the pig model and contribute to exploring the pathogenic mechanism of *G. parasuis* and the development of *G. parasuis* subunit vaccine.

## ACKNOWLEDGEMENTS

We thank Professor Liao and Professor Fan from the College of Veterinary Medicine, South China Agricultural University, Guang zhou, China for their kind provision of the integrative plasmid pSF116.

### Funding

This work was funded by the Science and Technology Strategic Cooperation Programs of Luzhou Municipal People's Government and Southwest Medical University (2019LZXNYDJ20) and University Sponsored Research Program of Southwest Medical University (2020ZRQNA010). The funding body supported preparation of test materials, and had no role in the design of the study and collection, analysis, and interpretation of data and in writing the manuscript.

### Grant Disclosures

The following grant information was disclosed by the authors:

The Science and Technology Strategic Cooperation Programs of Luzhou Municipal People's Government and Southwest Medical University: 2019LZXNYDJ20.
University Sponsered Research Program of Southwest Medical University: 2020ZRQNA010.

## Competing Interests

The authors declare there are no competing interests.

## Author Contributions

- Xuefeng Yan conceived and designed the experiments, performed the experiments, analyzed the data, prepared figures and/or tables, authored or reviewed drafts of the article, and approved the final draft.
- Ke Dai conceived and designed the experiments, performed the experiments, prepared figures and/or tables, and approved the final draft.
- Congwei Gu performed the experiments, prepared figures and/or tables, and approved the final draft.
- Zehui Yu performed the experiments, prepared figures and/or tables, and approved the final draft.
- Manli He analyzed the data, authored or reviewed drafts of the article, and approved the final draft.
- Wudian Xiao analyzed the data, authored or reviewed drafts of the article, and approved the final draft.
- Mingde Zhao analyzed the data, authored or reviewed drafts of the article, and approved the final draft.
- Lvqin He conceived and designed the experiments, performed the experiments, analyzed the data, prepared figures and/or tables, authored or reviewed drafts of the article, and approved the final draft.

## Animal Ethics

The following information was supplied relating to ethical approvals (i.e., approving body and any reference numbers):

Ethics committee of Southwest Medical University provided full approval for this research (20211017-001).

## Data Availability

The raw measurements are available in the Supplementary Files.

## Supplemental Information

Supplemental information for this article can be found online at http://dx.doi.org/10.7717/peerj.13648#supplemental-information.

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
