# Peer review of "Deletion of two-component system QseBC weakened virulence of Glaesserella parasuis in a murine acute infection model and adhesion to host cells"

_PeerJ, doi:10.7717/peerj.13648_

## Round 0.1 · original submission · Major Revisions

Dear authors,

Three independent referees completed the review report. Most criticism centers on the writing of the manuscript, particularly English grammar. Therefore, we highly recommend the authors invest in a good editor to re-write the manuscript so that it's in professional English.

The reviewers' reports follow, and we recommend that the authors attend to all, especially the significant comments, to improve this article.

Reviewer 1 ·

Basic reporting

The article is wordy and contains many grammar mistakes. The abstract has many spelling and grammar mistakes and should be reviewed. In the introduction, there is a whole section that is repeated (line 69 to 81 is identical to lines 59 to 69). Introduction should be reorganized and focused on the subject of the study.
Review M&M section and turn it more readable. Some section of M&M seems to be copied from laboratory protocols (mainly section 2.4, 2.6, 2.8 and 2.9) and are hard to follow. Therefore, the reader might get confused about what exactly has been done. These sections should be carefully rewritten.
Excess of information of equipment could be removed. The “natural transformation technique” indicated on line 119 and 124 should be described.
Results section should be shortened to indicate the results. Keep discussion for the discussion section.
Discussion.
Please start with your main finding and discuss the meaning of your data considering how they altered G. parasuis pathogenicity. Start at line 442. Everything prior to that (e.g lines 401 to 441) might be incorporated on your discussion as you go through your finding. Otherwise, most of it could be removed.

Experimental design

Satisfactory. M & M section could be improved.
This reviewer could not find the tables with the description of the primers used in the experiment.

Validity of the findings

Result are novel and add valuable information to current knowledge.

Additional comments

none

Reviewer 2 ·

Basic reporting

English needs to be improved. Many grammatical and spelling mistakes. Many sentences need proper reframing. Some sentences in the text are repetitive
The quality of the figures is bad. No resolution. Legend key needs to a full explanation of figure. Some figures does not reflect in terms of what authors report (microscopic images).

Experimental design

Methods need proper explanation
Some images quality is really bad
Some of them I don't see any difference what authors are saying

Validity of the findings

N/A

Additional comments

Authors need to analyze data properly. The quality of the figures is bad. Microscopic experiments need to be redone. I don't see any difference in the Wt vs mutant strains. Need to cite proper references. English needs to be improved, it does not read like publication quality.

·

Basic reporting

This is a very well designed project and have reported some very interesting findings on the role of QseBC in bacterial virulence. I like how the authors presented their findings in a step-wise manner and ensure the logics is correct.

However, the authors can use a lot of help on their language editing and proof reading. The manuscript's current writing has many grammatical flaws and the flow of some sections are not correct. I highly recommend the authors to invest in a good editor to re-write the manuscript so that it's in professional English.

One example:
the 3rd paragraph of Introduction is not well written: there is no intrinsic flow within the paragraph and content is repetitive (e.g., row 59-61 and 69-72, there are many more repetitive sentences in this paragraph)

Experimental design

Overall good and logical design.

Normally method section should be written in passive voice for reporting purposes. The manuscript currently has the method and approach written as instructions in many places. Please have a scientific editor to rewrite this section to ensure the readers are not confused with the methods

Validity of the findings

The findings are interesting and the authors provided both in vitro and in vivo data to support that QseBC is crucial for bacterial virulence.

However, as with many other sections, the result section needs to be re-written and properly proofread.

Two minor comments on the figure:
- Figure 4A: the lines are hard to read with the same color and similar pattern. Suggest using different color or add different shapes (solid circle, solid triangle. Empty triangle) to data points so that the distinction is clear
- For immunofluorescence images, I would recommend the author to increase the contrast so that the DAPI and CY3 staining comes out more clearly. Currently it is a bit challenging to see.

Additional comments

I want to emphasize that this manuscript has very high potential. Understanding the English is not the native language of the authors, I want to make sure that great academic work is not punished by linguistic limitations. Please have a professional editor rewrite the manuscript and ask a few international colleagues to quickly proofread before submitting again.

---

## Round 0.2 · Minor Revisions

Dear Dr. He,

The manuscript was significantly improved following the reviewers' suggestions.

However, I still detected some inconsistencies in MS that require a minor revision.

Please check final correction for theses sentences:

Line 71. Pseudomonas. Aeruginosa → Pseudomonas aeruginosa
Line 79. Ddeletion of qseC → Deletion of qseC
Line 106: to adhere to and invade cells → to adhere and invade cell.
Line 151: resistant transformants were identified identification by PCR → resistant transformants were identified by PCR
Line 201: amino acids(Invitrogen) → amino acids (Invitrogen)
Line 203: assays.Cells → assays. Cells
Lines 230-231 invasion,, and then vigorously washed three → invasion, and then vigorously washed three
Line 247: respectively.. The above cells were → respectively. The above cells were
Lines 248-250: abolish spaces
Line 255: The assay was performed three times in triplicates → The assay was performed in triplicates
Lines 282-284: The remaining groupfiveeight wasere allocated asto the PBS → The remaining group was allocated to the PBS
Line 285: or inability to eat or drink wereill be → or inability to eat or drink were ill be
Line 418: amplifyied → amplified
Line 441: non-polarity effect → no polar effect
Line 447: pPrimers → primers
Line 475: wasere observed → was observed
Line 507: AdhesionAdherence →Adhesion
Line 516: Decreased viruthe → Decreased virulence of the
Line 519: the survival rates was ere → the survival rates was
Line 528: howed the increased alveolar wall capillaries are severely congested !!!!
Line 608: further confirmed.This → further confirmed. This
Line 700: and PBMEC iswhich are the main cells inof the blood-brain barrier (BBB) !!!
Lines 703 and 707, 709, 726: parasuisHaemophilus parasuis !!!
Line 713: we speculatehypothesize !!!
Line 719: ais a global !!!
Lines 720-722: !!!!
Lines 729-730 !!!!
Line 730: wall wais !!!
Line 731: walls weare !!!!
Line 733: small numberamount !!!
Line 734: canobserved !!!
Line 735: wais !!!!
Line 736: wasare !!!!
Line 737: withand !!!!
Line 738: aroundsome
Line 739: ofsevere
Line 744: weare !!!!
Line 750: Theseis data !!!
Line 764: its abilitiesy !!!!

---

## Round 0.3 · accepted · Accept

After resolving all the points suggested, both major and minor, in the whole review process, I consider that the article is qualified to be published in PeerJ.